# Spatial Distribution of Noise Reduction in Four Iterative Reconstruction Algorithms in CT—A Technical Evaluation

**DOI:** 10.3390/diagnostics10090647

**Published:** 2020-08-28

**Authors:** Anette Guleng, Kirsten Bolstad, Ingvild Dalehaug, Silje Flatabø, Daniel Aadnevik, Helge E. S. Pettersen

**Affiliations:** 1Department of Oncology and Medical Physics, Haukeland University Hospital, 5021 Bergen, Norway; kirsten.bolstad@helse-bergen.no (K.B.); idaleh@ous-hf.no (I.D.); silje.flatabo@helse-bergen.no (S.F.); daniel.aadnevik@helse-bergen.no (D.A.); helge.egil.seime.pettersen@helse-bergen.no (H.E.S.P.); 2Department of Diagnostic Physics, Oslo University Hospital, 0424 Oslo, Norway

**Keywords:** computed tomography, iterative reconstruction, noise, anthropomorphic phantom, image analysis, inter-image standard deviation, CT vendor comparison

## Abstract

Iterative reconstruction (IR) is a computed tomgraphy (CT) reconstruction algorithm aiming at improving image quality by reducing noise in the image. During this process, IR also changes the noise properties in the images. To assess how IR algorithms from four vendors affect the noise properties in CT images, an anthropomorphic phantom was scanned and images reconstructed with filtered back projection (FBP), and a medium and high level of IR. Each image acquisition was performed 30 times at the same slice position, to create noise maps showing the inter-image pixel standard deviation through the 30 images. We observed that IR changed the noise properties in the CT images by reducing noise more in homogeneous areas than at anatomical edges between structures of different densities. This difference increased with increasing IR level, and with increasing difference in density between two adjacent structures. Each vendor’s IR algorithm showed slightly different noise reduction properties in how much noise was reduced at different positions in the phantom. Users need to be aware of these differences when working with optimization of protocols using IR across scanners from different vendors.

## 1. Introduction

Since computed tomography (CT) was introduced in the 1970s, it has become an indispensable tool in diagnostic medicine. Over the years, revolutionizing technological advancements to the modality have provided improved image quality, and allowed for more diverse clinical applications. While a CT scan gives life saving diagnostic information, there is also a small risk associated with the X-ray radiation exposure involved in the procedure [1,2]. For this reason, several technological advancements have focused on reducing the radiation dose necessary to obtain the desired image quality, like improvements in detector technology, the use of automatic tube current modulation and improvements in image reconstruction algorithms [3].

Iterative reconstruction (IR) methods were used to reconstruct images already in the first clincial CTs [4]. As image matrix sizes increased and spatial resolution improved, filtered back projection (FBP) was developed to provide shorter reconstruction times [5]. IR methods were re-introduced into daily clinical routine as faster computer technology made it practical to use, and today all CT vendors have developed their own IR algorithms. They are often used in addition to conventional FBP reconstruction. The IR algorithms aim at reducing image noise to provide an improved image quality at the same, or lower, radiation dose level compared to FBP, and reduce artefacts specific to FBP reconstruction. This is achieved by using a combination backward reconstructions, where images are reconstructed from real and virtual projection data, and forward reconstructions, where virtual projection data are generated from the current image [4,6]. The image acquisition process is simulated to different degrees of sophistication, depending on the available computational power and time constraints [4]. For each iteration of the algorithm, the obtained data are optimized in either the raw data domain, the image domain or both [6].

Noise magnitude in a CT image can be defined as the standard deviation of pixel values in an otherwise homogeneous area of the image. Since all vendors have developed their own IR algorithms, there will be differences in where and how much each algorithm reduces noise in the image. The purpose of this study was to investigate the noise properties of four IR algorithms, one from each of the main CT vendors, by quantifying and comparing the amount of noise reduced at and outside anatomical edges in the image. Solomon and Samei [7] have previously investigated how the use of one IR algorithm affects the spatial distribution of noise in phantom images where anatomical edges are present. Our study extends their work by including four IR algorithms from different vendors, which to our knowledge has not been attempted in any previous studies. We also introduce a new method for quantifying and comparing the amount of noise reduced at and outside anatomical edges in the image, which is better suited for the phantom used in this study.

IR can also change the frequency distribution of noise, and thus the visual appearance of noise in the image. The effect of IR on the frequency distribution of noise in images of both homgeneous and anthropomorphic phantoms has been documented in previous studies [7,8,9,10,11], and will not be investigated further in this study.

An anthropomorphic phantom was scanned and images reconstructed using FBP, a medium level of IR, and a high level of IR. The noise reduction across the images were evaluated by creating inter-image pixel standard deviation maps, i.e., noise maps, and inter-image pixel noise reduction maps, i.e., noise reduction maps, enabling quantification and subjective evaluation of the differences in noise reduction properties between the algorithms.

## 2. Materials and Methods

In this study, the anthropomorphic PH-5 CT-Abdomen Phantom (Kyoto-Kagaku, Kyoto, Japan) was scanned with a Chest-Abdomen-Pelvis protocol on CT scanners from all main CT-vendors: Canon Aquilion Prime (Canon Medical Systems, Otawara, Tochigi, Japan), GE Revolution Evo (GE Healthcare, Waukesha, WI, USA), Philips Ingenuity (Philips Healthcare, Cleveland, OH, USA) and Siemens Somatom Definition Flash (Siemens Healthineers, Forchheim, Germany). The phantom is shown in Figure 1.

The image acquisition was based on scan- and reconstruction parameters recommended by the American Association of Physicists in Medicine (AAPM) for Chest-Abdomen-Pelvis protocols on each scanner [12]. For the GE Revolution Evo scanner, no AAPM protocol recommendations were available, so scan- and reconstruction parameters were based on recommendations for the GE Discovery scanner. A fixed tube current was used, adjusted to give a similar volume CT dose index (CTDI${}_{\mathrm{vol}}$) between 12 mGy and 15 mGy across all vendors. The obtained images were reconstructed with filtered back projection (FBP), and a medium and high level of iterative reconstruction (IR). All scan- and reconstruction parameters are listed in Table 1.

To assess the noise properties of the IR algorithms, each image acquisition was performed 30 times at the same slice position in the phantom with identical scan- and reconstruction settings. This gave a set of 30 identical images, except for quantum noise fluctuations, electronic noise and potential phantom displacement. By calculating the inter-image standard deviation in each pixel position across the 30 images, creating a noise map, the spatial distribution of noise in the phantom, with a one-pixel resolution, can be visually assessed [7,8,9,13].

The amount of noise reduction due to IR compared to FBP was measured at different positions in the phantom. Three anatomical edges were chosen, with differences in pixel intensities over the respective edges being 1000 HU, 70 HU and 30 HU. The edges are shown in Figure 1b. Edge profiles showing the amount of noise present at and outside each edge was measured in the noise maps over a line ROI with a width of 10 pixels positioned perpendicular to each anatomical edge, averaging pixel values along the ROI width. All measurements were performed using the ImageJ (U. S. National Institutes of Health, Bethesda, MD, USA) software [14].

To quantify the average noise reduction both at the anatomical edges and in the homogeneous areas outside the edges, noise reduction maps were created, showing the pixelwise relative noise reduction between two noise maps. Each IR level was compared to FBP by using the formula [7]
$$NRM=\frac{NM_{FBP}-NM_{IR}}{NM_{FBP}},$$
where NRM is the noise reduction map being calculated, $NM_{FBP}$ is a noise map created from FBP reconstructed images, and $NM_{IR}$ is a noise map created from images reconstructed with a given level of IR. The noise reduction maps can be seen in Appendix A. The same line ROIs as shown in Figure 1b was used to measure noise reduction over each edge in the noise reduction maps. Average noise reduction at each edge was found by averaging the measured pixel values at the position of the given edge. Similarly, the average noise reduction outside each edge was found by averaging the measured pixel values on both sides of the given edge.

## 3. Results

### 3.1. Noise Maps

Figure 2 shows the noise maps for FBP and two levels of IR for all scanners. IR reduces noise, but does not preserve the spatial noise distribution in the CT images. In other words, when using IR more noise is reduced in homogeneous areas and less at anatomical edges, making the spatial distribution of the noise more heterogeneous compared to FBP. As the IR level increases, so does the difference in noise reduction between homogeneous areas and edges.

### 3.2. Noise Profiles across Edges

Figure 3, Figure 4 and Figure 5 shows the noise profiles measured over the 1000 HU, 70 HU and 30 HU edges respectively in the noise maps for all scanners. For all three edges and all vendors, the noise is reduced more outside the edge than at the edge when using IR. As the contrast over the edge decreases, i.e., reduced difference in CT numbers between the two adjacent organs, the amount of noise reduced at and outside the edge becomes more similar. The amount of noise reduced at and outside the three edges for all scanners and IR levels, as measured in the noise reduction maps in Appendix A, are presented in Table 2.

## 4. Discussion

### 4.1. Noise Reduction Properties of the IR Algorithms

For the GE, Philips and Siemens scanners, images reconstructed with FBP showed more noise in the center of the phantom, and less towards the phantom periphery, both in the horizontal and vertical direction (see Figure 2). The same tendency was seen when applying IR: the noise in the homogeneous areas of the phantom still showed more noise in the phantom center, and less towards the phantom periphery. This is thought to be due to the increased scatter contributions to the central regions of the phantom, caused by an increased likelihood for Compton scatter interactions where the traversed path length is longest. IR algorithms use a regularization term to model and reduce noise [15], a method which is well-suited to reduce scattered radiation since we can predict where it is expected to appear in the image. This shows that these IR algorithms reduce noise in the image while preserving the relative noise distribution in homogeneous areas.

For the Canon scanner, reconstruction with FBP showed more noise in a horizontal band through the phantom center (see Figure 2), which has been documented in previous studies by Merzan et al. [16] and Guleng [13]. This is believed to be due to a softer X-ray beam, leading to a higher attenuation along the horizontal direction. When applying IR, this horizontal band disappears, and the difference in noise between the center and periphery of the phantom is small. This shows that for the Canon scanner, applying IR on the CT image results in a more even distribution of noise in homogeneous areas compared to FBP.

In accordance with previous studies by Solomon and Samei, Gervaise et al., Silva et al. and Noël et al., all IR algorithms were shown to reduce noise compared to FBP, with an increase in noise reduction for an increase in IR level [7,17,18,19]. The Canon scanner showed the smallest increase in noise reduction for increasing IR level: noise reduction outside the 1000 HU edge increased by 5 percentage points (pp) when using a high level of IR compared to a medium level, while the GE, Philips and Siemens scanner showed an increase of approximately 20 pp. Reconstruction with a medium and high level of IR on the Canon scanner will therefore provide images with a more similar noise magnitude compared to the other scanners. This does not necessarily mean that the medium and high IR Canon images will look more similar: A higher level of IR will also have an effect on the noise power spectrum (NPS), and thus the visual appearance of noise in the image [10], which has not been concidered in this study.

Furthermore, in accordance with previous studies, by Solomon and Samei, Funama et al. and Dalehaug et al., the noise reduction was found to be more prominent in homogeneous areas of the phantom compared to the noise reduction at anatomical edges [7,8,9]. This implies that the IR algorithms are less aggressive with the removal of noise in the presence of high gradients in the image, in order to preserve edges and small objects. The amount of noise reduced at a certain edge is dependent on the difference in density between the two adjacent structures. The noise reduction increases as the difference in density decreases, i.e., more noise is reduced at the 30 HU edge than at the 1000 HU edge. The IR algorithms thus preserves high contrast edges better than it preserves low contrast edges. This can be seen for all scanners: The difference between noise reduced at and outside a given edge decreases with decreasing contrast over the edge. For low contrast edges, the Canon scanner was shown to have the smallest difference in noise reduced at and outside the edge.

Based on the work presented in this study, it is evident that there are differences between the output of the IR algorithms provided by Canon, GE, Philips and Siemens. It is useful to be aware of these differences when working with optimization of protocols using IR across different scanners. Still, further work needs to be done to assess the clinical implications of these findings. The anthropomorphic phantom used in this study, while mimicking anatomical structures in the body, is not representative for the real-life differences in density, texture and anatomical details in the human body. The organs in the phantom have sharp, well defined edges and a homogeneous density. In a real patient, the organ and tissue edges will be less sharp, with larger variations in density inside a given organ. Some IR behavior will therefore be impossible to evaluate with this phantom, e.g., how the algorithm handles a more gradual change in tissue density. It is hard to replicate this study on patients due to the excess dose, and to the requirement of identical objects throughout the scans. However, Funama et al. have applied this methodology on porcine tissue with comparable results [8].

Another potential limitation of this study, is the lack of a multireader evaluation of the phantom images. Even if an anthropomorphic phantom is a simplified representation of the complex human anatomy, an experienced radiologist might be able to notice tendencies in the phantom images that could be indicative of the IR algorithms’ clinical performance. However, the algorithms’ performance as observed in simple phantoms will not fully represent how they will perform in clinical images.

### 4.2. Scan and Reconstruction Parameters

There are many parameters that affect the noise level and the spatial distribution of noise in a CT image. While this study compared FBP to two levels of IR for four different IR algorithms, parameters like dose and reconstruction kernel were kept unchanged. Previous studies by Löve et al. [10] and Dalehaug et al. [9] have shown that a dose reduction of respectively 90% and 80% can have a significant effect on the noise reduction properties of IR algorithms. The choice of a different reconstruction kernel could change the noise magnitude and texture in the image. If the differences between two kernels is large enough, this could affect the IR behavior. To further investigate the noise reduction properties of the four IR algorithms, it would therefore be possible to scan with a significantly lower dose level, e.g., a dose corresponding to a low dose protocol, and other reconstruction kernels, e.g., a sharp lung kernel.

The CTDI${}_{\mathrm{vol}}$ used in this study was chosen to be similar to a standard dose for a Chest-Abdomen-Pelvis scan. The dose on the Philips scanner was slightly lower than for the other scanners (12 mGy compared to 15 mGy), which will produce slightly more noise in the images. This could affect the noise reduction when applying IR. Still, both 12 mGy and 15 mGy can be considered a normal dose for a Chest-Abdomen-Pelvis scan, and the tendencies observed for the noise properties of Philips’ IR algorithm is not expected to be significantly affected by this discrepancy in dose. This is supported by Löve et al. [10], where a 30% dose reduction was shown to provide a similar noise reduction for increasing IR level for all vendors.

The amount of noise in a CT image is strongly dependent on the reconstruction kernel used. In this study, a soft kernel was used on all scanners, chosen in accordance with recommendations from AAPM. The soft kernels will not be identical, and will have slightly different effects on the image noise. A previous study by Solomon, Christianson and Samei [20], comparing the noise texture correspondence between GE and Siemens kernels in FBP images, show that the kernels used in our study (the GE “Standard” and Siemens “B30f” kernels) provide similar noise textures, but does not constitute a best match. It would therefore be possible to choose kernels that provides a more similar noise texture between these two scanners. Since there is not necessarily a one-to-one correspondence between the best matching kernels on two scanners, it is not likely that four different kernels from four scanners all will constitute a best match with each other. It was therefore deemed acceptable to use kernels that provided a good noise texture correspondence, even if they did not constitute the best match available. As for the kernels used on the Canon and Siemens scanner, kernel matching performed with the same method [20] showed that the Canon “F18” and Siemens “B30f” kernels both show the best noise texture correspondence to each other. We believe that the use of soft kernels recommended by AAPM all will provide similar noise texture in the images, as has been shown to be the case for GE, Siemens and Canon.

### 4.3. Method for Measuring Noise Reduction

To our knowledge, no previous studies have attempted to quantify the amount of noise reduced at and outside anatomical edges when using IR on scanners from different vendors, and there is no standardized way to perform such noise reduction measurements. This study has utilized a simple and intuitive analytical method, suited to perform noise reduction measurements in images with the same amount of scattered radiation present as in an adult-sized anthropomorphic phantom. The method can only be applied to well defined edges surrounded by homogeneous material, and three suitable edges were examined. More information about IR behavior could be uncovered by investigating edges in other positions of the phantom, and with other differences in density between the adjacent tissues.

When subtracting the average of the 30 FBP images obtained on one scanner from each of the individual images, some structures in the phantom were still visible. This may indicate that some movement was present in the phantom between the images, even if all 30 images were set to image identical slice positions. Some of this shift in position can be explained by movement of the patient table. For some scanners the table had to be re-positioned between each obtained image, and inaccuracies in the table positioning system or small movements in the phantom could be causing the shifts. A similar, but smaller, shift in phantom position was seen even for the scanners where there was no table movement between images. This could be caused by vibrations in the gantry translating or rotating the phantom slightly. The movement could contribute to a larger standard deviation in pixels positioned around edges in the noise maps. To assess the impact of these small variations in position, an analysis was performed in MICE Toolkit 1.1.3 (NONPI Medical AB, Umeå, Sweden) using a rigid Elastix tranformation between image number 1 and 15, and between image number 1 and 30 to find the total translation between each pair of images. The average translation for all vendors was found to be 0.02 mm (with a maximal translation of 0.03 mm), which, with a pixel width of 0.6–0.7 mm, gives a sub-pixel difference in position between the sampled images. This, combined with the fact that we are not observing an increase in noise along edges in the FBP reconstructed images in the same way as we do for IR, shows that the shifts in phantom position are small enough that they will not alter the tendencies observed in this study.

## 5. Conclusions

Iterative reconstruction changes the spatial distribution of noise in a CT image. Noise is reduced more in homogeneous areas compared to anatomical edges. This tendency is more visible at increasing IR levels, and for increasing differences in density at the edge between the two adjacent structures. Each vendor’s IR algorithm also shows slightly different noise reduction properties in how much noise is reduced at different positions in the phantom. The results are expected to be indicative of performance in clinical images, and users need to be aware of these differences when working with optimization of protocols using IR across scanners from different vendors.

## Figures and Tables

**Figure 1 diagnostics-10-00647-f001:**
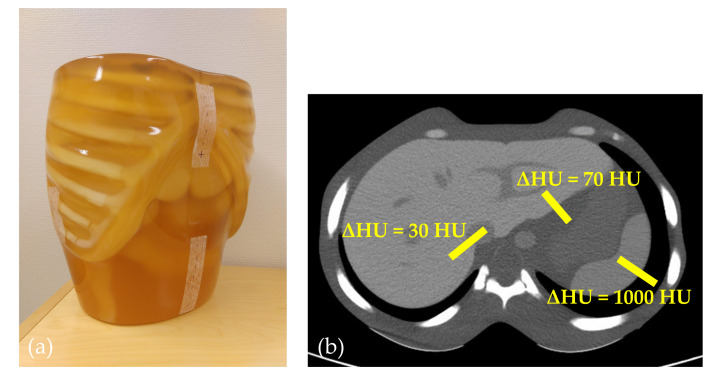
(**a**) Anthropomorphic abdomen phantom; (**b**) computed tomography (CT) image of the phantom illustrating the positions of the three edges used to measure noise profiles in the noise maps. The difference in tissue density over the three edges were 1000 HU, 70 HU and 30 HU.

**Figure 2 diagnostics-10-00647-f002:**
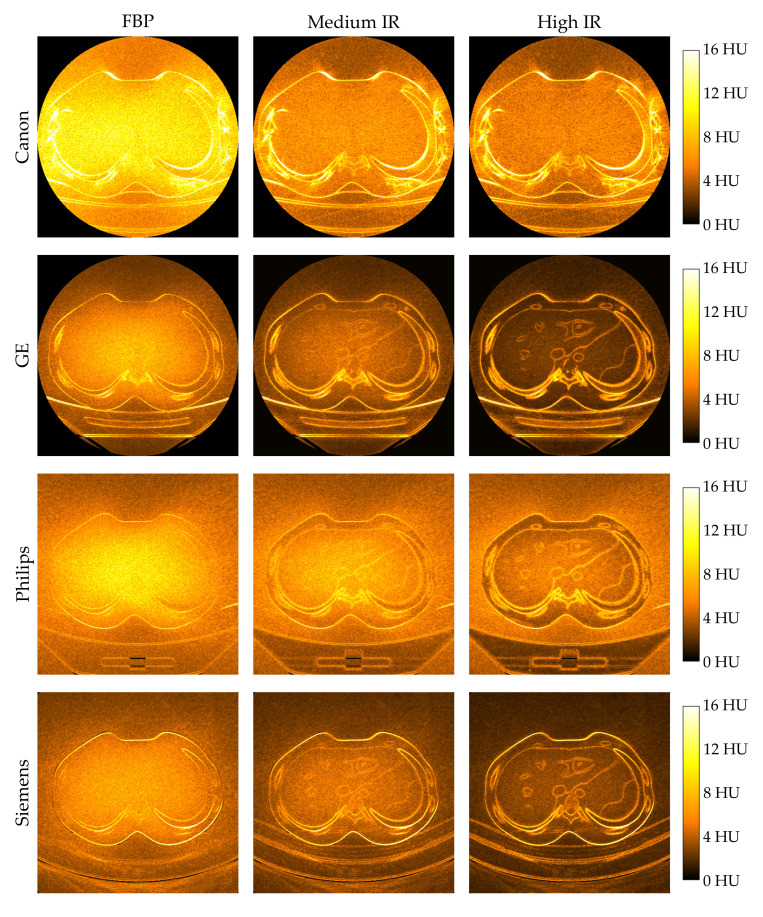
Noise maps showing the inter-image pixel standard deviation for 30 images, each reconstructed with filtered back projection (FBP), a medium level of iterative reconstruction (IR) and a high level of IR for all vendors. A lighter color indicates a higher standard deviation, an thus a higher level of noise, in the given pixel.

**Figure 3 diagnostics-10-00647-f003:**
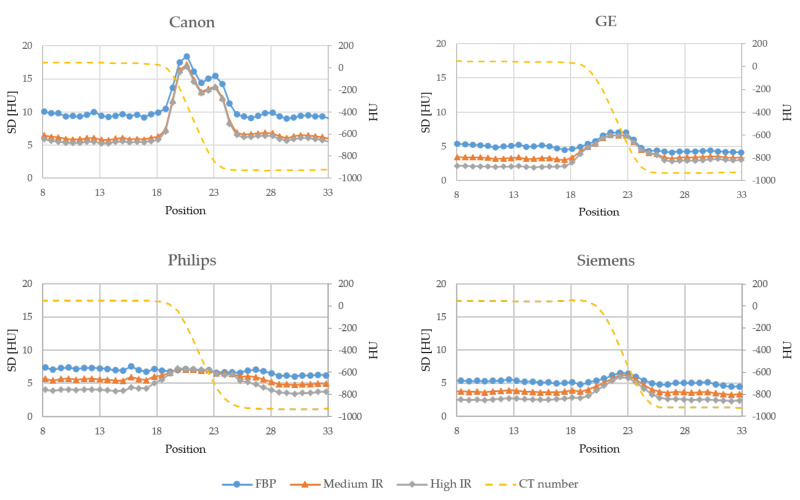
Edge profiles showing the standard deviation (SD) measured over the 1000 HU edge in the noise maps reconstructed with FBP, a medium level of IR and a high level of IR for each vendor. The dashed line shows the average CT number in the FBP images.

**Figure 4 diagnostics-10-00647-f004:**
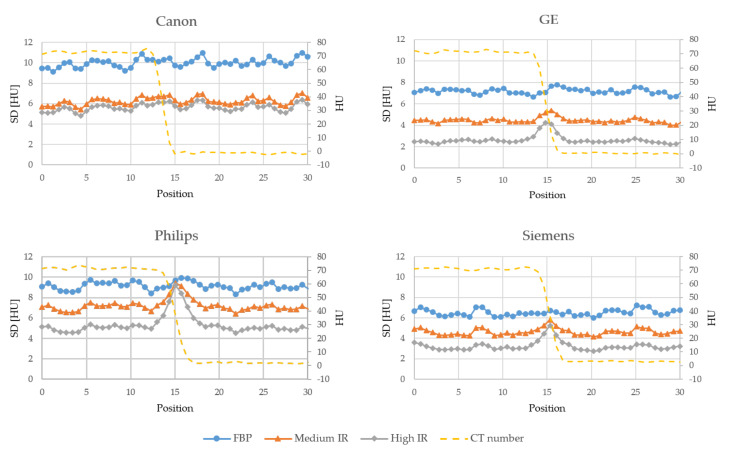
Edge profiles showing the standard deviation (SD) measured over the 70 HU edge in the noise maps reconstructed with FBP, a medium level of IR and a high level of IR for each vendor. The dashed line shows the average CT number in the FBP images.

**Figure 5 diagnostics-10-00647-f005:**
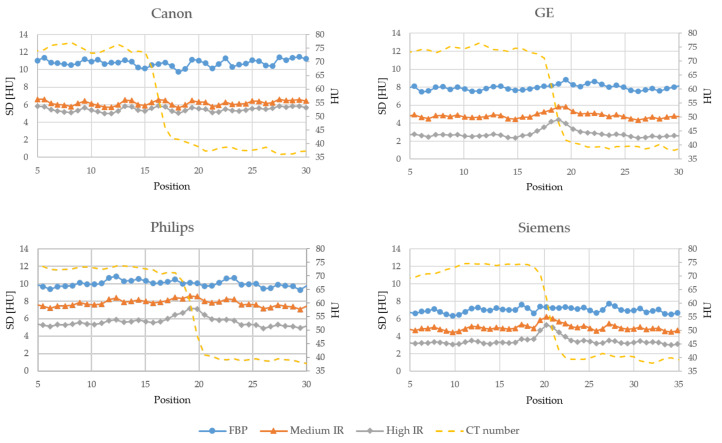
Edge profiles showing the standard deviation (SD) measured over the 30 HU edge in the noise maps reconstructed with FBP, a medium level of IR and a high level of IR for each vendor. The dashed line shows the average CT number in the FBP images.

**Table 1 diagnostics-10-00647-t001:** Scan- and reconstruction parameters for all vendors.

Parameter	Canon	GE	Philips	Siemens
Scanner type	Aquilion Prime	Revolution Evo	Ingenuity	Somatom Definition Flash
CTDI${}_{\mathrm{vol}}$ [mGy]	15	15	12	15
Tube potential [kV]	120	120	120	120
Tube current product [mAs]	160	250	150	222
Reconstruction kernel	FC18	Standard	B	B30f / I30f *
IR algorithm	AIDR 3D	ASIR-V	iDose	ADMIRE
IR levels used	org/std/str	0%/50%/100%	0/3/6	0/3/5

CTDI${}_{\mathrm{vol}}$, volume computed tomography dose index; AIDR 3D, Adaptive Iterative Dose Reduction 3D; org/std/str, original/standard/strong; ASIR-V, Adaptive Statistical Iterative Reconstruction—Veo; ADMIRE, Advanced Modeled Iterative Reconstruction. * B30f used with FBP, I30f with IR.

**Table 2 diagnostics-10-00647-t002:** Average amount and standard deviation of noise reduced outside and at three anatomical edges for two levels of IR relative to FBP, measured in the calculated noise maps (see Appendix A). The difference in noise reduced outside and at each anatomical edge is listed in percentage points (pp).

Edge	IR Level	Position	Noise Reduction Relative to FBP			
			Canon	GE	Philips	Siemens
1000 HU	Medium	Outside edge	36% ± 3%	31% ± 8%	22% ± 1%	29% ± 2%
		At edge	8% ± 2%	6% ± 0.2%	0% ± 1%	8% ± 3%
		Difference	(28 ± 3) pp	(25 ± 8) pp	(22 ± 1) pp	(21 ± 4) pp
	High	Outside edge	41% ± 4%	51% ± 15%	43% ± 2%	52% ± 2%
		At edge	10% ± 1%	6% ± 0.4%	0% ± 2%	15% ± 6%
		Difference	(31 ± 4) pp	(45 ± 15) pp	(44 ± 3) pp	(37 ± 6) pp
70 HU	Medium	Outside edge	38% ± 2%	39% ± 1%	23% ± 0.4%	30% ± 1%
		At edge	34% ± 1%	29% ± 2%	7% ± 3%	20% ± 5%
		Difference	(4 ± 2) pp	(10 ± 2) pp	(16 ± 3) pp	(10 ± 5) pp
	High	Outside edge	44% ± 2%	65% ± 1%	46% ± 1%	53% ± 1%
		At edge	40% ± 1%	44% ± 4%	12% ± 6%	34% ± 9%
		Difference	(4 ± 2) pp	(21 ± 4) pp	(33 ± 6) pp	(18 ± 9) pp
30 HU	Medium	Outside edge	42% ± 2%	40% ± 1%	23% ± 1%	30% ± 1%
		At edge	40% ± 1%	33% ± 2%	16% ± 2%	18% ± 3%
		Difference	(2 ± 2) pp	(7 ± 2) pp	(7 ± 2) pp	(11 ± 3) pp
	High	Outside edge	49% ± 2%	67% ± 1%	46% ± 1%	52% ± 1%
		At edge	46% ± 1%	52% ± 4%	32% ± 3%	33% ± 5%
		Difference	(3 ± 2) pp	(15 ± 4) pp	(14 ± 3) pp	(19 ± 5) pp

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
