# Peer review of "Spatial Distribution of Noise Reduction in Four Iterative Reconstruction Algorithms in CT—A Technical Evaluation"

_diagnostics, 2020, doi:10.3390/diagnostics10090647_

Round 1

Reviewer 1 Report

Spatial distribution of noise reduction is important, so does spatial frequency distribution of noise. I suggest the authors to perform 2D or 3D NPS measurements for the four CT scanners. Please describe the novelty of the manuscript. What are the new contributions of the authors and their work from Solomon, J. (refs 8 & 19). 

Author Response

Dear reviewer,

The authors would like to wholeheartedly thank you for your responses to this work, which to our opinion has improved its quality. Below are answers to the raised concerns. In addition to the changes described below, minor spelling errors have been corrected throughout the manuscript.

Spatial distribution of noise reduction is important, so does spatial frequency distribution of noise. I suggest the authors to perform 2D or 3D NPS measurements for the four CT scanners.

The focus of this study has been the spatial distribution of noise reduction, especially across edges of various contrast levels. The authors agree that also the spatial frequency distribution of noise from the studied IR algorithms is of great interest. However, we expect that calculating NPS across edges (which is the scope of this study) yield little information as this is usually the domain of MTF which has been compared in (Andersen 10.1016/j.ejro.2018.02.002).  Taking the NPS of homogeneous areas resulting from the studied IR algorithms has been performed elsewhere (Löve 10.1259/bjr.20130388; Andersen 10.1016/j.ejro.2018.02.002), while the “anthropomorphic” NPS resulting from subtraction of multiple acquisitions can be found in (Solomon 10.1118/1.4893497; Funama 10.1016/j.ejmp.2014.02.005). As such, we consider NPS calculations to be redundant with the published results and the scope of the study in mind. Line 49-52 has been added in the Introduction to highlight this.

Please describe the novelty of the manuscript. What are the new contributions of the authors and their work from Solomon, J. (refs 8 & 19).

The spatial distribution of noise reduction when using IR compared to FBP has previously been investigated by Solomon and Samei for one IR algorithm (Solomon 10.1118/1.489349). Our study extends this work by including one IR algorithm from each of the four main CT vendors. This allows us to compare their behavior, and has to our knowledge not been done in any previous studies. We also introduce a new method for quantifying the amount of noise reduced at and outside anatomical edges, which is better suited for larger phantoms where more scattered radiation is present. The Introduction, line 43-48, and Discussion, line 200-201, has been revised to make this more clear.

Reviewer 2 Report

This manuscript provides several insights on the noise behavior of different levels of IR algorithms distributed by different CT vendors. However, it seems hard to see which could be the practical impact of the study findings into radiologists' practice, i.e. how and to what extent they could affect perceived image quality in specific types of CT examinations. I would therefore suggest performing an additional multireader evaluation of visual image quality from anthropomorphic phantoms along with objective measurements of image parameters.

Author Response

Dear reviewer,

The authors would like to wholeheartedly thank you for your responses to this work, which to our opinion has improved its quality. Below are answers to the raised concerns. In addition to the changes described below, minor spelling errors have been corrected throughout the manuscript.

This manuscript provides several insights on the noise behavior of different levels of IR algorithms distributed by different CT vendors. However, it seems hard to see which could be the practical impact of the study findings into radiologists' practice, i.e. how and to what extent they could affect perceived image quality in specific types of CT examinations. I would therefore suggest performing an additional multireader evaluation of visual image quality from anthropomorphic phantoms…

The focus of this study was to perform a technical evaluation of the spatial distribution of noise reduction, especially across edges of various contrast levels, when using IR algorithms from four different vendors. The results from this study alone are not sufficient to fully characterize the impact from IR on clinical image quality, and the algorithm’s performances as observed in simple phantoms will not fully represent how they will perform in complex clinical images. The authors fully agree that a multireader evaluation would be of great interest. However, to keep the study concise and focused the authors have therefore considered a multireader evaluation to be out of scope for this study. The title of the manuscript has been revised to make the focus of the study more clear, and now reads: “Spatial Distribution of Noise Reduction in Four Iterative Reconstruction Algorithms in CT – A Technical Evaluation”.

…along with objective measurements of image parameters.

Thank you for addressing this. The comparison of objective image parameters like CT-number, noise magnitude, NPS, CNR and spatial resolution for IR algorithms from different vendors has been performed by, among others, (Löve 10.1259/bjr.20130388; Andersen 10.1016/j.ejro.2018.02.002). To our knowledge no previous study has investigated and compared the spatial distribution of noise between IR algorithms from four vendors. To keep the study concise and focused it was chosen not to include other objective measurements of image parameters in our investigation. Line 49-52 has been added in the introduction, with references to previous studies, to highlight this.

Round 2

Reviewer 2 Report

Thank you for your reply. I would still suggest commenting on the lack of a multireader evaluation of image quality as a potential study limitation at the end of the discussion section, as per your response ("The results from this study alone are not sufficient to fully characterize the impact from IR on clinical image quality, and the algorithm’s performances as observed in simple phantoms will not fully represent how they will perform in complex clinical images").

Author Response

Dear reviewer,

The authors would like to thank you again for your valuable feedback to this work, which to our opinion continues to improve its quality. Below is our response to the raised concern.

Thank you for your reply. I would still suggest commenting on the lack of a multireader evaluation of image quality as a potential study limitation at the end of the discussion section, as per your response ("The results from this study alone are not sufficient to fully characterize the impact from IR on clinical image quality, and the algorithm’s performances as observed in simple phantoms will not fully represent how they will perform in complex clinical images").

We have added line 162-166 in the Discussion section, describing the lack of a multireader evaluation of image quality as a potential limitation of this study.